# CausalGame: Benchmarking Causal Thinking of LLM Agents in Games

**Zhenhao Chen**[*1] **Yongqiang Chen**[*1,2] **Chenxi Liu**[*3] **Junchi Yu**[4] **Xiangchen Song**[2]
[1]MBZUAI [2]Carnegie Mellon University [3]TMLR Group, Hong Kong Baptist University

**Zijian Li**[1,2] **Jialin Li**[5] **Philip Torr**[4] **Bo Han**[3] **Kun Zhang**[1,2]
[4]University of Oxford [5]New York University, Abu Dhabi
zhenhao.chen@mbzuai.ac.ae  yqchen24@gmail.com  cscxliu@comp.hkbu.edu.hk

## Abstract

Recently, it has received growing attention in building AI Scientist agents with Large Language Models (LLMs). Since scientific discovery fundamentally relies on uncovering causal relationships from observations, the capability of *causal thinking* that distinguish causation from correlation and hidden biases, is essential to LLM agents. Despite a number of existing benchmarks for AI scientists, none of them are designed with the consideration of hidden biases and confounders, that widely exist in real-world scientific discovery. To this end, we present `CausalGame`, a benchmark that evaluates the causal thinking capabilities of LLM agents through interactive games. More specifically, we ask LLM agents to actively design experimental protocols, collect observation data and derive a final solution with an explanation report. To emulate realistic scientific discovery challenges, we design 14 game settings with the incorporation of selection bias, noisy measurements, and hidden confounders. The results with 16 frontier LLM agents show that they consistently fail to reason about and recover the underlying causal relationships required to solve the games. `CausalGame` provides a rigorous assessment of capabilities essential to AI Scientist agents.

## 1 Introduction

As the large language models (LLMs) demonstrate increasing capabilities in reasoning and resolving complex tasks (Guo et al., 2025; Li et al., 2025; Plaat et al., 2025), it has sparked growing curiosity and discussion in the community to build LLM-based AI Scientist agents to automate the process of scientific discovery (ZHENG et al., 2025; Zhou et al., 2025). LLMs are shown to demonstrate great promise in automating a number of research tasks, such as conducting literature surveys (Lu et al., 2024), proposing useful hypotheses (Mitchener et al., 2025), writing papers Yamada et al. (2025), and improving experimental codes for open-ended discovery (Novikov et al., 2025; Lange et al., 2025).

On the other hand, science originates from identifying critical variables and revealing the underlying causal mechanisms (Hanson, 1958; Kuhn & Hawkins, 1963; Wallace, 1981; Spirtes et al., 2000). Causality and causal thinking are essential to distinguishing statistical correlations from the causal relations and to establishing rigorous scientific conclusions (Glymour; Pearl, 2009). For example, the existence of hidden confounders and selection bias can mislead the conclusions driven by statistics (Doll & Hill, 1950; Simpson, 1951). Shown as in Fig. 1, if we were using an AI Scientist agent in medicine and the agent is incapable of causal thinking, then the agent can give incorrect treatment that may cause a severe issue. Despite the necessity of causal thinking in scientific discovery, it has been surprisingly neglected in developing and measuring the capabilities of AI Scientists. As LLM agents are at the core of existing AI Scientists, hence, we ask an intriguing research question:

*Are existing LLM agents capable of causal thinking?*

Although there exists a number of benchmarks specifically designed for AI Scientists, they mostly focus on execution of the scientific research pipeline (Wan et al., 2026; Liu et al., 2025), statistical-

---

*These authors contributed equally.

driven data analysis (Chan et al., 2024; Jing et al., 2024), or law discovery through regression upon *observed* variables (Shojaee et al., 2025; Zheng et al., 2025), none of them considers the challenges imposed by *hidden variables*. Nevertheless, the awareness of the hidden variables are critical in bringing a number of scientific breakthroughs (Glymour; Wallace, 1981).

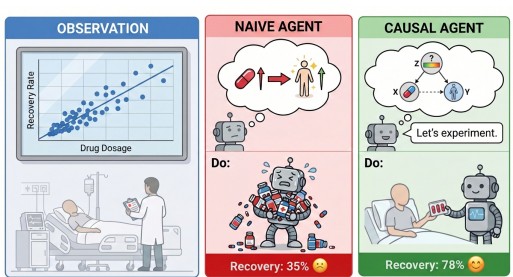

Therefore, we present a benchmark, `CausalGame`, that simulates the real-world scientific discovery practice into a number of game scenarios, where the agent is required to interact with the environment, collect data and observations, design and perform experiments, analyze the data, and draw conclusions. Hence, in `CausalGame`, the objective of the agent is to properly design drones, e.g., determining the attributes of the important components of drones. Those drones will be dispatched to execute tasks and suffer from severe weather conditions or attacks by enemies. The relations between the vulnerability of drone components, the weather conditions, and the enemy attacks are characterized by an underlying *structural causal model* (SCM). The agent will have some

Figure 1: The necessity of causal thinking for AI scientists. Observed correlations (left) can be misleading due to hidden confounders. Unlike a naive agent that treats correlation as causation (middle), a causal agent identifies underlying mechanisms through active experimentation (right).

budgets to send small patches of the drones to collect the data and gain an understanding of the underlying *causal process*. The understanding will be reported and used to come up with the final design of the drones. Like the real-world scientific discovery, we will evaluate both the quality of the drone design and the report through rubrics.

More importantly, the flexible design in `CausalGame` allows us to incorporate realistic challenges in real-world scientific discovery. Specifically, we construct several game scenarios to incorporate the selection bias, noisy measurements, and hidden confounders (Spirtes et al., 2000). For example, the agent can only observe *survived* drones throughout the turns. Agents that are limited in the capability of causal thinking can easily be biased and suffer from spurious statistical clues. Even when the agent can obtain high ratings by luck, the evaluation design of `CausalGame` can easily distinguish those incapable of causal thinking.

The final outcome of `CausalGame` consists of 14 game scenarios. Through comprehensive evaluation of 16 frontier LLMs, including the state-of-the-art reasoning LLMs that are widely used for building AI Scientist agents, we find that LLM agents are lack of causal thinking, are easily misled by statistical clues, and fail to uncover the underlying causal mechanism of the environment. Those results highlight the intrinsic drawbacks of LLMs in building AI Scientist agents.

## 2  RELATED WORK

**AI Scientist Agents.** Recent advances in LLM-based agents have drawn increasing attention to the concept of AI scientists, which has great potential for accelerating scientific discovery Lu et al. (2024); Yamada et al. (2025). The goal of AI scientists is to automate core components of the scientific workflow, including literature review Huang et al. (2025b), hypothesis generation Yang et al. (2025), and the systematic design and evaluation of experiments Huang et al. (2025a). Early efforts in this direction focused on building general AI Scientist frameworks with broad research ability Gottweis et al. (2025). Recent work has shifted towards viewing the AI scientist framework as a cognitive layer Zhang et al. (2025a) of scientific research by integrating domain-specific knowledge, specialized tool sets, and *in silico* simulation Wang et al. (2025a). This paradigm has made huge achievements in biomedicine Swanson et al. (2025); Truhn et al. (2026), earth science Feng et al. (2025), material science Ghafarollahi & Buehler (2025), computer science Novikov et al. (2025), and chemistry Yang et al. (2026); Boiko et al. (2023). To further bridge the gap between the dry-lab research and wet-lab validation, recent studies also explore the integration of embodied AI and robotics for wet-lab automation Zhu et al. (2022); Tom et al. (2024), enabling closed-loop scientific discovery.

**Benchmark for Scientific Capability** Faithfully benchmarking the scientific capability of LLMs and LLM-based agents is becoming imperative as they are the foundation for AI scientists. Early

Table 1: Comparison to the existing benchmarks. `CausalGame` features in automated evaluation, imitating real-world data-driven, multi-turn scientific discovery, fine-grained evaluation of the explanation provided by the agent, and the challenges raised by the underlying hidden causal variables.

| Paradigm | Dataset | Automated evaluation | Data-Driven | Multi-turn interaction | Causal relations | Fine-grained verification | Hidden variables |
|---|---|---|---|---|---|---|---|
| Scientific Research | Zhang et al. (2025b) | ✓ | ✗ | ✗ | ✗ | ✓ | ✗ |
| | Liu et al. (2025) | ✓ | ✗ | ✗ | ✗ | ✓ | ✗ |
| | Starace et al. (2025) | ✓ | ✗ | ✗ | ✗ | ✓ | ✗ |
| Data Science | Chan et al. (2024) | ✓ | ✓ | ✓ | ✗ | ✗ | ✗ |
| | Jing et al. (2024) | ✓ | ✓ | ✓ | ✗ | ✗ | ✗ |
| | Majumder et al. (2024) | ✓ | ✓ | ✗ | ✗ | ✗ | ✗ |
| | Verma et al. (2025) | ✓ | ✓ | ✗ | ✓ | ✗ | ✗ |
| | Wang et al. (2025c) | ✓ | ✓ | ✓ | ✗ | ✓ | ✗ |
| Others | Zheng et al. (2025) | ✓ | ✓ | ✗ | ✓ | ✗ | ✗ |
| | Mandal et al. (2025) | ✗ | ✓ | ✗ | ✗ | ✗ | ✗ |
| | Swanson et al. (2025) | ✗ | ✓ | ✓ | ✓ | ✗ | ✗ |
| | Roohani et al. (2024) | ✗ | ✗ | ✓ | ✓ | ✗ | ✗ |
| | CausalGame | ✓ | ✓ | ✓ | ✓ | ✓ | ✓ |

studies, such as MMMU Yue et al. (2024) and ScienceQA Lu et al. (2022), focused on benchmarking the general scientific knowledge of LLMs via multi-modal and multi-disciplinary scientific question answering (QA) Yue et al. (2025); Hu et al. (2025); Rein et al. (2024). Later benchmarks curated more specialized and advanced scientific QA tasks to benchmark advanced scientific understanding Wan et al. (2026); Li et al. (2026); Yu et al. (2025); Phan et al. (2025b). Recent works aim to benchmark the capability of LLM-based agents in the workflow of scientific discovery, rather than scientific QA tasks. These benchmarks evaluate agentic capabilities across different stages of research, including the ideation Liu et al. (2025); Si et al. (2025), review synthesis Zhang et al. (2025b), data analysis Wang et al. (2025b); Shojaee et al. (2025), coding Starace et al. (2025), scientific law discovery Zheng et al. (2025), and experiment design Mandal et al. (2025).

**Scientific discovery** ultimately seeks *causal* and *mechanistic* knowledge—claims about how a system would change under interventions and why—rather than correlations that hold only under a fixed data-generating process Pearl (2009); Woodward (2004). In practice, causal discovery is complicated by latent confounding, selection effects, and measurement noise, which can make observational regularities misleading and can render causal directions unidentifiable without targeted interventions Spirtes & Glymour (1991). Accordingly, a substantial literature treats discovery as an *active* process, asking which interventions and sequences of experiments most efficiently identify causal structure Hyttinen et al. (2013). In many scientific domains—especially physics and dynamical systems—causal structure is tightly coupled to identifying governing dynamics from trajectories Yao et al. (2024). These perspectives motivate interactive evaluations where an agent must design experiments and reason causally under confounding/bias/noise to recover underlying mechanisms.

The key differences between `CausalGame` and the existing works are given in Table 1. Although existing benchmarks provide a holistic evaluation of LLM-based AI Scientists, they lack in replicating real-world scientific discovery, which is usually data-driven, and the agent needs to design experiments and interact with the environments to collect more observations to draw scientific conclusions. Moreover, the evaluation of the scientific report is also essential as it provides the explanation of the discovered causal mechanism. The closest benchmarks related to `CausalGame` are Acharya et al. (2025); Verma et al. (2025) that also benchmark the capabilities of LLMs in doing causal inference from the data science perspective. Nevertheless, they lack in the replication of the real-world scientific discovery and the consideration of challenges raised by the hidden variables in causality.

# 3 CAUSALGAME BENCHMARK

## 3.1 BASIC GAME SETTING

The basic game setting of `CausalGame` is multi-turn interaction between agents and the environments: agents actively propose experimental actions, collect observations, and iteratively refine their strategies. Concretely, each turn deploys a batch of drones with a chosen design (allocating defense across components) under a sampled environment, and returns outcome

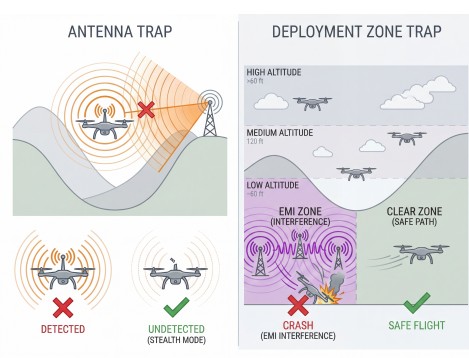

Figure 2: Illustration of game scenarios in `CausalGame`.

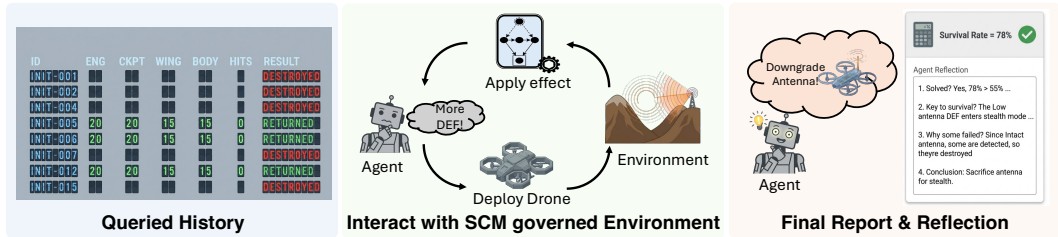

**Figure 3:** Illustration of `CausalGame` pipeline. The agent is given some historical records of surviving drones and required to interact with the environment to gain an understanding of the underlying causal mechanism.

signals together with partial observation. To mimic real-world discovery constraints, the feedback is incomplete (some internal variables are hidden) and the observed data can be biased or noisy. Finally, the agent is evaluated by a one-shot submission on a large fleet after a budgeted exploration phase.

**Game scenarios.** `CausalGame` contains a suite of scenarios instantiated from a common simulator. As shown in Fig. 2, we start from two base scenarios: `Antenna Trap`, where a latent weather pattern affects both antenna damage and detection risk, and a surviving antenna can increase radar detection via hidden signal emission (thus "protecting the antenna" can be suboptimal); and `Deployment Zone Trap`, where an unobserved mission zone jointly determines the deployment corridor (e.g., altitude band) and the true failure driver (e.g., EMI), inducing a strong but spurious correlation that can mislead correlational strategies. Each scenario is specified by a configuration file (`game.json`) that sets resource budgets, the agent-visible interface, and scenario parameters.

**SCM as the game engine.** Our core design choice is to treat the SCM as the scenario "engine": scientific discovery aims to uncover the hidden data-generating mechanism, and in `CausalGame` this mechanism is exactly the SCM behind each scenario. Concretely, each scenario corresponds to an SCM that specifies structural equations over variables $X_1, \ldots, X_d$ with exogenous noise $U_i$,

$$X_i := f_i(\mathrm{Pa}(X_i), U_i), \quad i = 1, \ldots, d, \tag{1}$$

which induces the environment sampling and the downstream outcomes returned to the agent.

**Agent interactions.** At the beginning of the game, the agent can access some historical observations to gain a basic understanding of the game. Then, at each turn, the agent interacts with the environment by analyzing the existing results or designing new drones and dispatches the drones for testing. After the agent consumes all the budgets, the agent is asked to provide a final design of the drone.

**Final Report.** After the exploration phase, the agent submits a single final design that is evaluated on a large fleet (Stage 2). We report success using the fleet survival rate, and define a *win* if it exceeds a scenario-specific threshold, which is set to be approximately 5%–8% below the theoretical optimal survival rate for that scenario. In addition to the numeric score, agents are required to produce a short natural-language report that explains their final design choice based on the evidence collected

## 3.2 CAUSAL THINKING.

Causal thinking is to infer how and why an outcome would change under hypothetical interventions, rather than merely describing associations observed in data (Spirtes et al., 2000). Across the history of science, many influential findings were initially obscured or misinterpreted due to systematic biases arising from data collection, measurement processes, or unobserved common causes (Wallace, 1981; Glymour). These challenges motivated the development of causal approaches that go beyond correlational analysis. Table 2 lists some representative historical scientific discovery cases that illustrate three recurring obstacles in causal inference and their underlying mechanisms.

**Selection bias** arises when the process by which data are selected depends on variables related to both variables, inducing spurious dependencies in the observed data. This phenomenon is exemplified by Berkson's demonstration of spurious correlations in hospital-based populations Berkson (1946) and Sackett's empirical analysis of admission rate bias in case–control studies Sackett (1979).

In this benchmark, we introduce controlled selection biases: the agent can only observe *survived* drones. For example, in `Antenna Trap`, the antenna can be destroyed by bad weather, while

Table 2: A summary of causal mechanisms and representative historical cases.

| Challenge | Causal Mechanism | Historical Cases |
|---|---|---|
| Selection bias | $X \rightarrow S \leftarrow Y$ | • Spurious correlation induced by hospital-based selection Berkson (1946)
• Admission rate bias in hospital case–control studies Sackett (1979) |
| Measurement error | $X \leftarrow Z \rightarrow Y$, with $X_{\text{obs}}$, $Z_{\text{obs}}$, $Y_{\text{obs}}$ | • Severe attenuation of disease risk estimates due to dietary measurement error Kipnis et al. (2003)
• Regression calibration to correct dietary measurement error in postmenopausal breast cancer studies Prentice et al. (2013) |
| Latent confounders | $X \leftarrow U \rightarrow Y$ | • Common-cause explanation for the smoking–lung cancer association Doll & Hill (1950)
• Unmeasured smoking as a latent confounder in radiation–lung cancer cohort studies Richardson et al. (2014) |

drones with a destroyed antenna will be less likely to be detected by enemies and have more chance to survive. Hence, the agent will observe a majority of *surviving drones with damaged antennas*. A natural mitigation is to strengthen the antenna, which suffers from the spurious statistical clues. In addition, we can also strengthen the survival biases in historical data.

**Measurement error** arises when latent variables of interest are imperfectly observed through noisy proxies. Dietary epidemiology provides clear evidence of this issue, where the OPEN biomarker study showed severe attenuation of disease risk estimates due to dietary measurement error Kipnis et al. (2003), and regression calibration was shown to partially recover associations in postmenopausal breast cancer studies Prentice et al. (2013). In this benchmark, we inject noises to measurements with varying magnitudes to evaluate whether LLMs can reason robustly under observational imperfections.

**Latent confounders** arise when unobserved common causes jointly influence two variables, generating non-causal associations. Classic and modern examples include the common-cause hypothesis in the smoking–lung cancer debate Doll & Hill (1950) and the demonstration that unmeasured smoking substantially confounds radiation–lung cancer associations in occupational cohorts Richardson et al. (2014). Determining the causal relations requires revealing the latent confounders underlying the observed variables. In this benchmark, by initially withholding critical variables, we test whether LLMs can be aware of the potentially existing latent confounders and actively propose what additional variables should be observed through interacting with environments. Moreover, one could also inject the spurious correlations caused by the latent confounders in the historical data.

## 3.3 RUBRIC-BASED EVALUATION

While survival rate provides a quantitative measure of task performance, it alone cannot distinguish *why* agents fail or how they reason about the causal structure. To address this, we develop a fine-grained rubric evaluation framework that assesses agent behavior along four complementary dimensions. The explicit criteria are summarized in Table 16. Each agent session is evaluated by an LLM-based judge using this rubric, yielding dimension-wise scores and an overall rubric score.

**Causal Reasoning (11 points)** evaluates whether the agent correctly identifies the core causal mechanisms specified in the Task Report, avoids known traps or spurious correlations, and provides mechanistic explanations with sufficient depth. High-scoring responses must articulate explicit causal chains (rather than correlations), include intermediate variables or processes, and propose testable predictions or validation strategies.

**Experimental Design (2 points)** assesses whether the agent supports its conclusions with concrete experimental evidence. Agents are expected to cite specific numerical results (e.g., survival percentages or controlled comparisons) and clearly connect these results to the claims.

**Reflection Quality (2 points)** evaluates the agent's ability to reflect on its own reasoning and outcomes. High-quality reflections should identify concrete errors, blind spots, or unverified assumptions that are directly traceable to the proposed approach, rather than offering vague or generic caveats.

**Data Usage (1 point)** examines whether the agent explicitly links observed data to its conclusions. Agents must state which specific data, comparisons, or measurements support each claim, avoiding unsupported or purely speculative reasoning.

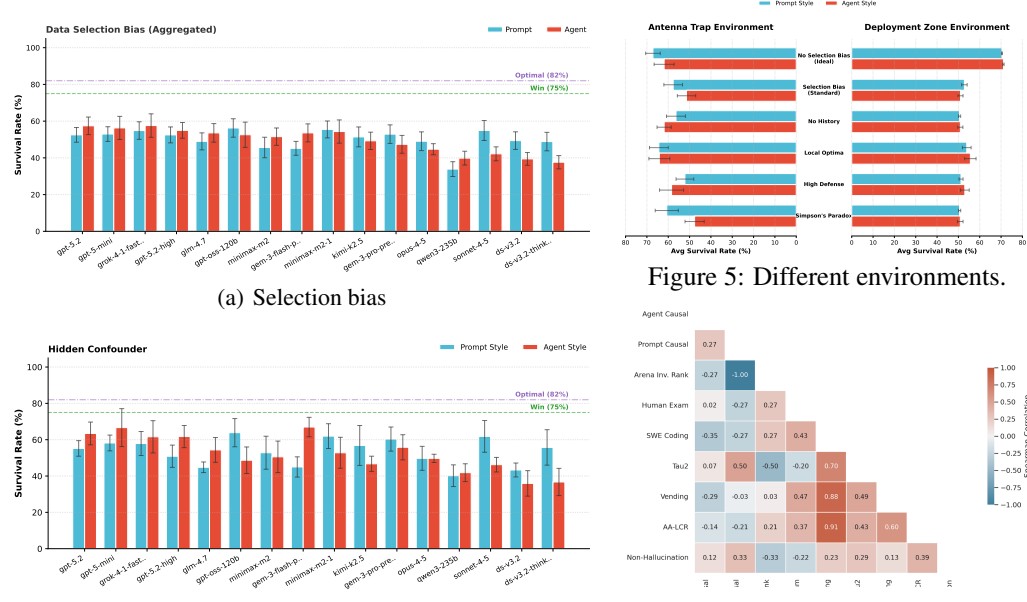

(a) Selection bias

Figure 5: Different environments.

(b) Hidden confounders

Figure 4: Main results of agents in `CausalGame`.

Figure 6: Benchmarks Correlation

## 4 EXPERIMENTS

**Experimental settings.** In experiments, we select 16 frontier LLMs that have shown impressive performance across various challenging reasoning and agentic tasks, including GPT-5.2, GPT-5.2-high thinking, GPT-5-mini, Grok-4-1-fast reasoning, GLM-4.7, GPT-OSS-120b, MiniMax-M2, MiniMax-M2-1, Gemini-3-Flash, Gemini-3-Pro, Kimi-k2.5, Claude-Opus-4.5, Claude-Sonnet-4.5, Qwen3-235B-A22b-thinking, Qwen3-235B-A22b-thinking, DeepSeek-v3.2, and DeepSeek-v3.2-thinking. As LLMs are generically good at multi-turn reasoning, we consider both direct prompting and agentic evaluation settings via the ReAct framework (Yao et al., 2023).

**Observation 1: State-of-the-art frontier LLMs fail to think with causality.** The main results are given in Fig. 4, where we aggregate and average the performance of different LLMs under selection bias and hidden confounders. We also draw two lines, where one is the winning condition we set for the game, and the other is the optimal performance. From the results, we can find that all of the frontier LLMs fail to win the game by performing causal thinking.

**Observation 2: Efficient LLMs can perform competitively or even surpass large LLMs.** Interestingly, the distilled LLMs perform competitively or even surpass the full counterpart LLMs, such as GPT-5-mini vs GPT-5.2, or Gemini-3-Flash vs Gemini-3-Pro. One possible explanation could be that the powerful reasoning capabilities of large LLMs make them more concentrated on statistical clues instead of causation.

**Observation 3: Agentic framework does not necessarily bring improvements.** Moreover, interestingly, when evaluated in an agentic framework, the reasoning LLMs can even underperform the counterparts evaluated in direct prompting, such as Claude-4.5-Opus. It suggests promising future investigations in training reasoning agents with causal thinking capability.

**Observation 4: LLMs have certain capabilities in designing useful experiments.** Fig. 5 presents the averaged performance of all the LLM agents under different history data setting. Under ideal settings or the "No history" setting, LLMs require probing the environment by designing experiments, and obtain certain success. More interestingly, given the "Local optimal" setting where a local optimal design is given to the LLM, many of the LLMs are able to iterate over the local optimal design and find better solutions. However, when the biases increase or the action space grows more complicated (from `Antenna Trap` to `Deployment Zone Trap`), the failure persists.

**Observation 5: None of the existing benchmarks examine the causal thinking capability.** Fig. 6 reports the Spearman rank correlations between causal-thinking results and a range of existing agentic benchmark, ranging from hallucination (Chiang et al., 2024), reasoning (Phan et al., 2025a), coding (Jimenez et al., 2023), long-horizon reasoning (Barres et al., 2025; Backlund & Petersson, 2025), long-context understanding (Team, 2025), and parametric knowledge (Jackson et al., 2025).

Table 3: Five failure-mode patterns.

| Pattern | Code | Definition |
|---|---|---|
| Complete Collapse | A | No evidence of *Causal Reasoning* (Causal Reasoning = 0) and no systematic *Experimental Design* (Experimental Design < 1.0). |
| Experimental Attempt | B | Demonstrates *Experimental Design* ($\geq$ 1.0) but fails to identify any core causal mechanism (Causal Reasoning = 0). |
| High–Low Capability | C | High rubric score ($\geq$ 9.0) with strong *Reflection Quality*, yet low survival rate (< 50%). |
| Data-Driven | D | Lacks *Causal Reasoning* but leverages *Data Usage* from deployment outcomes. |
| Weak Progress | E | Minimal but consistent progress across all rubric dimensions. |

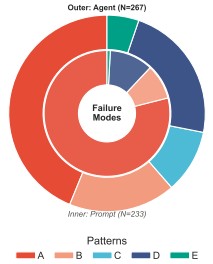

Figure 7: Failure mode

Table 4: Hypothesis Analysis (Antenna Trap)

| Model | Golden | Distractor |
|---|---|---|
| Grok-4.1-Reasoning | 75% | 72% |
| DeepSeek-V3.2-Exp-Thinking | 50% | 40% |
| GPT-5.2 | 75% | 79% |
| Qwen3-235B | 60% | 60% |
| Kimi-K2.5 | 65% | 80% |

Table 5: Hypothesis Analysis (Zone Trap)

| Model | Golden | Distractor |
|---|---|---|
| GPT-5.2 | 41% | 46% |
| DeepSeek-V3.2-Exp-Thinking | 31% | 46% |
| Kimi-K2.5 | 38% | 79% |
| Qwen3-235B | 22% | 67% |
| Grok-4.1-Reasoning | 19% | 83% |

Table 6: Selection Bias on Survival Rate

| Condition | Prompt | Agent |
|---|---|---|
| w Selection Bias | 55.2 | 51.2 |
| w/o Selection Bias | 68.8 | 66.5 |
| $\Delta$ | **+13.5** | **+15.3** |

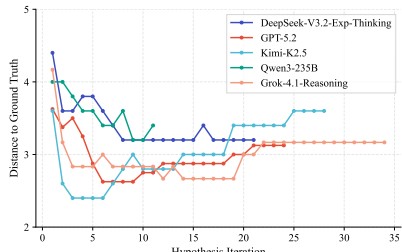

Figure 8: Distance Trajectories (Antenna Trap)

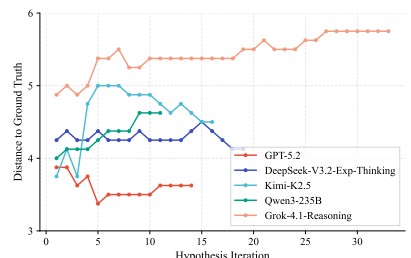

Figure 9: Distance Trajectories (Zone Trap)

We can find that the causal thinking capabilities in both agentic and prompting settings only weakly correlate with other capabilities benchmarks. Moreover, we observe a clear structural separation between *agentic style* and *prompt style* causal thinking results. Specifically, both the results from agent causal thinking and prompt causal thinking are positively correlated with Non-Hallucination Rate and tau-2 benchmark, where the latter evaluates the multi-turn tool call capabilities. Particularly, prompt causal thinking has much stronger correlations with the compared benchmarks than agent causal thinking. Interestingly, both causal thinking styles are negatively correlated with coding SWE. One plausible explanation is that coding tasks primarily reward pattern completion and syntactic correctness, whereas causal thinking requires hypothesis formation, verification, and reflection, which may involve different reasoning processes.

**Failure mode analysis.** To gain a fine-grained understanding of the failures modes of LLM agents in `CausalGame`, we analyze agent behaviors by jointly considering rubric scores and task performance (i.e., survival rate), which yields five qualitatively distinct failure-mode patterns (Table 3). These patterns characterize and taxonomize *how* the agents fail. Pattern A (*Complete Collapse*) reflects a total lack of causal engagement. Pattern B (*Experimental Attempt*) indicates procedural experimentation without causal inference. Pattern C (*High–Low Capability*) combines strong rubric performance and reflection with poor survival. Pattern D (*Data-Driven*) relies on observational deployment data in the absence of causal reasoning. Pattern E (*Weak Progress*) shows limited but consistent capability across all rubric dimensions.

**Agent Style.** Agent Style exhibits substantial behavioral diversity (Fig. 7, outer ring): **43.8%** Pattern A, **17.6%** Pattern B, **10.5%** Pattern C, **22.8%** Pattern D, and **5.2%** Pattern E. Notably, **Pattern C emerges as a distinct and non-trivial failure mode** (28/267 trails): agents achieve high rubric scores ($\geq$ 9/22) and strong reflection quality ($\geq$ 1.0/2.0), yet suffer from low survival (mean 27.3%).

This highlights the necessity of combining both rubric-based and score-based evaluations to provide a holistic view of agentic capabilities.

Fig. 10 shows that each pattern exhibits a distinct and internally consistent score profile (rather than noise-driven artifacts). Pattern A remains uniformly weak across dimensions, with only modest *Data Usage* (around the low-20% range). Patterns B/D/E/C all achieve similar *Experimental Design* scores (roughly ∼20%), but diverge sharply thereafter: Pattern C and Pattern E peak in *Reflection Quality* (both around ∼50%), consistent with the "high-reflection" characterization of Pattern C despite low survival; Pattern D and Pattern E attain the highest *Data Usage* (both around ∼50%), whereas Pattern B and Pattern C are lower (roughly mid-20% and high-30%, respectively). Across all patterns, *Causal Reasoning* remains near-zero to low-single-digit percentages, indicating key limitations of existing LLMs lying in the limited causal reasoning capabilities.

**Prompt Style** shows a sharply skewed distribution (Fig. 7, inner ring): **79.0%** Pattern A, **9.0%** Pattern B, **11.2%** Pattern D, with **no instances of Pattern C** and only **0.9%** Pattern E. Mean rubric scores are substantially lower (0.24–2.96/22) than Agent Style (3.00–7.40/22). The absence of Pattern C indicates that single-turn prompting rarely produces deceptively convincing but causally hollow reasoning; failures in Prompt Style are more transparent, but exploration is severely limited.

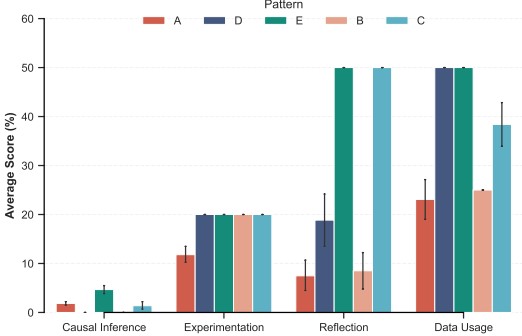

Figure 10: Rubric scores by failure mode.

**Comparison and implications.** Agent Style enables diverse behavioral strategies—experimental (B), data-driven (D), reflective (C), and partially balanced (E)—whereas Prompt Style collapses almost entirely to Pattern A. The exclusive emergence of Pattern C in Agent Style highlights a subtle but important risk: agents can *appear* competent under rubric evaluation while failing in real task execution. This suggests that evaluation protocols for agentic systems should emphasize *behavior-grounded outcomes* in addition to self-reflection and explanation quality.

**Hypothesis Analysis.** To quantify the hypothesis quality, we decompose the ground-truth hypothesis into a set of atomic claims in Antenna- and Zone- Trap tasks. We also decompose several distractor claims in each task. See details Tab. 15 in Appendix. During the iteration process, we calculate the distance between agent's current hypothesis with the ground truth as $\#\{golden\ claims\ not\ found\} + \#\{distractor\ claims\ found\}$. As shown in Fig. 8-9 and Tab. 4-5, the selected models differ in tasks. Grok-4.1 is the winner in Antenna Trap while being the worst in Zone Trap; GPT-5.2 is competitive in both tasks. Trajectories present significant non-monotonic pattern in all models, indicating the limited ability to effectively improve hypothesis.

**Impact of Selection Bias.** Table 6 shows the impact of selection bias on model performance. Removing selection bias through balanced sampling yields substantial improvements: +13.5% for prompt mode and +15.3% for agent mode. Notably, agent mode exhibits greater sensitivity to selection bias. We hypothesize this stems from the compounding nature of sequential decisions. Early biased observations lead to biased deployments, further generating biased data, progressively reinforcing spurious correlations across turns. In contrast, prompt mode processes all observations simultaneously, partially mitigating bias amplification.

## 5    CONCLUSIONS

In this work, we constructed a benchmark `CausalGame` that instantiated the challenges of real-world scientific discovery in 14 game scenarios, including selection bias, noisy measurement, and hidden confounders. Our benchmarking with 16 frontier LLMs through both direct prompting or agentic framework shows that they all fail to design proper experiments to uncover the underlying causal mechanism of the environment. We also present a detailed rubric-based analysis and demonstrate that the key attribution of their failures is the lack of causal thinking capability. The results highlight the necessity of incorporating causal thinking in building AI Scientist agents.

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

## LLM Use Statement

From the research side, this work studies benchmarking LLM agents' capabilities in causal thinking through the construction of games. From the paper writing side, we use LLMs to assist with improving the writing of this work.

## Ethics Statement

We study benchmarking the LLM agents' capabilities using games. The results provide a clear assessment of the LLM agent's capabilities in automating scientific discovery, which will benefit the whole humanity and society. This work does not involve human subjects or personally identifiable information beyond public benchmarks used under their licenses.

## A  Details of CausalGame Benchmark

This appendix provides comprehensive details about the CausalGame benchmark, including execution modes, scenario descriptions, and the prompts used for evaluation.

### A.1  Execution Modes

CausalGame supports two execution modes that represent different paradigms for LLM-based agent interaction: **Agentic (Hybrid)** mode using structured tool calling, and **Prompting (Legacy)** mode using code execution.

Table 7: Comparison of Execution Modes in CausalGame

| Aspect | Agentic (Hybrid) | Prompting (Legacy) |
|---|---|---|
| API Operations | Structured Tool Calling | Python code with `client.xxx()` |
| Data Analysis | Sandboxed Python execution | Full code execution |
| Control Flow | Explicit tool calls with reasoning | Code blocks with API calls |
| Reasoning | Mandatory ReAct pattern | Optional |
| Exploration Guard | Must deploy before submit | None |
| Tool Limit | Max 5 tools per turn | Unlimited API calls |

**Agentic (Hybrid) Mode.**  In this mode, the agent interacts with the environment through structured function calling. The agent must explicitly invoke tools such as `get_status`, `get_history`, `deploy_drone`, and `submit_final_design`. Each turn requires the agent to provide reasoning before taking actions, following the ReAct (Reasoning and Acting) paradigm. A key safety feature is the exploration guard: the agent must deploy at least one drone before submitting a final design, preventing premature submissions without data collection.

**Prompting (Legacy) Mode.**  In this mode, a pre-configured `client` object is injected into the Python execution namespace. The agent writes Python code blocks that directly call methods like `client.deploy_drone()` and `client.get_history()`. This mode allows for more flexible data analysis through unrestricted code execution but lacks the structured reasoning requirements of the agentic mode.

### A.2  Benchmark Scenarios

CausalGame includes 14 carefully designed scenarios organized into three families, each presenting distinct causal reasoning challenges. Table 8 summarizes all scenarios with their causal challenges.

#### A.2.1  Antenna Trap Family

The Antenna Trap scenarios are inspired by real-world signal detection problems where a functioning communication system can paradoxically increase risk.

Table 8: Overview of All CausalGame Scenarios

| Experiment | Selection Bias | Hidden Confounder | Threshold |
|---|---|---|---|
| *Antenna Trap Family* | | | |
| antenna_trap | ✓ | – | 75% |
| antenna_trap_high_def | ✓ | ✓ | 75% |
| antenna_trap_local_optima | ✓ | – | 75% |
| antenna_trap_no_history | ✓ | – | 75% |
| antenna_trap_no_selection_bias | – | – | 75% |
| antenna_trap_simpsons_paradox | ✓ | ✓ | 75% |
| *Deployment Zone Trap Family (Farr's Cholera Paradox)* | | | |
| deployment_zone_trap_categorical | ✓ | – | 75% |
| deployment_zone_trap_categorical_high_def | ✓ | ✓ | 75% |
| deployment_zone_trap_categorical_local_optima | ✓ | – | 75% |
| deployment_zone_trap_categorical_no_history | ✓ | – | 75% |
| deployment_zone_trap_categorical_no_selection_bias | – | – | 75% |
| deployment_zone_trap_categorical_simpsons_paradox | ✓ | ✓ | 75% |
| deployment_zone_trap_env_shift | ✓ | – | 75% |
| *Weather Family* | | | |
| weather_noise | ✓ | – | 55% |

**Causal Structure.** The underlying causal graph contains the following relationships:

- Weather $\rightarrow$ Wind Intensity $\rightarrow$ Antenna Damage
- Antenna HP $\rightarrow$ Signal Emission $\rightarrow$ Detection Probability $\rightarrow$ Combat Engagement
- Combat Engagement $\rightarrow$ Drone Damage $\rightarrow$ Survival

**The Trap.** Historical data shows that drones with higher antenna DEF (defense) values tend to survive better in the training distribution. This creates a spurious correlation: agents naturally conclude that maximizing antenna DEF improves survival. However, the true causal mechanism is that a *functional* antenna emits signals that increase detection probability by enemy systems, leading to combat and destruction. The optimal strategy is to set `antenna_def=0`, allowing storms to destroy the antenna early, which activates "stealth mode" and dramatically reduces detection.

**Variants.**

- **high_def**: Adds pressure to allocate high DEF values, creating an additional confounder.
- **local_optima**: Introduces local optima that trap gradient-following strategies.
- **no_history**: Removes historical flight data, requiring pure exploration.
- **no_selection_bias**: Control condition without selection bias.
- **simpsons_paradox**: Data exhibits Simpson's paradox where aggregate trends reverse within subgroups.

### A.2.2 DEPLOYMENT ZONE TRAP FAMILY

This family is inspired by Farr's Cholera Paradox, a historical example where altitude appeared to protect against cholera when the true cause was water source contamination at lower elevations.

**Causal Structure.**

- Deployment Zone $\rightarrow$ Altitude (Visible)
- Deployment Zone $\rightarrow$ EMI Level (Hidden) $\rightarrow$ Communication Failure
- Communication Failure $\rightarrow$ Mission Failure $\rightarrow$ Drone Loss

**The Trap.**   Agents observe that low-altitude flights have significantly higher loss rates and may conclude that engine upgrades (for altitude capability) are the solution. However, the true causal factor is electromagnetic interference (EMI), which is hidden from initial observations. Low-altitude zones happen to have high EMI levels, creating the spurious altitude-survival correlation. The optimal strategy is to maximize `shield_def` for EMI protection and select the `signal_filter` enhancement module.

**Enhancement Modules (Categorical Variant).**   The categorical variant requires agents to select one enhancement module:

- `radar_boost`: No EMI protection (trap)
- `thermal_shield`: No EMI protection (trap)
- `power_core`: No EMI protection (trap)
- `stealth_coating`: No EMI protection (trap)
- `signal_filter`: Provides 55% EMI reduction (optimal)

**Variants.**

- **high_def**: Additional DEF allocation pressure.
- **local_optima**: Local optima traps.
- **no_history**: No historical data available.
- **no_selection_bias**: Control condition.
- **simpsons_paradox**: Simpson's paradox in aggregated data.
- **env_shift**: Distribution shift between exploration and validation phases.

### A.2.3   WEATHER FAMILY

The Weather family scenarios test the agent's ability to handle environment-dependent effects and noisy observations.

**Weather Noise.**   This scenario adds observation noise that varies with weather conditions:

- Rainy conditions: 20% observation noise
- Clear conditions: 5% observation noise

Agents must distinguish genuine causal patterns from noise artifacts and deploy sufficient samples to average out the noise, particularly in rainy conditions. The victory threshold is lowered to 55% to account for increased difficulty.

### A.3   PROMPTS AND TOOL DEFINITIONS

### A.3.1   LEGACY MODE PROMPT

The following prompt template is used for Legacy (Prompting) mode execution:

Listing 1: Legacy Mode Prompt Template

```
# Antenna Trap Experiment - Agent Prompt

## ROLE
You are an advanced Drone Designer working on a drone optimization project.
Your goal is to optimize drone designs for survival in a hostile canyon
environment. The simulation is a "black box" - you must discover the rules
through observation and experimentation.

## KEY INFORMATION
- You control **DEF (armor/defense)** values for each drone component
- HP (health points) are FIXED and HIDDEN - you cannot see or modify them
- **VALID design keys** (use EXACTLY these names):
  - `engine_def`, `cockpit_def`, `wing_def`, `body_def`
```

```
  - `antenna_def`, `camera_def`, `gun_def`
- Different environmental conditions may affect drone performance
- **Not all environmental data is visible!** Use `client.query_environment()`
  to discover hidden measurements

## INVESTIGATION STRATEGY
1. **OBSERVE**: Look for patterns in survival, hit_count, and environmental
   conditions
2. **DISCOVER**: Query for hidden measurements (weather, atmospheric, etc.)
3. **ANALYZE**: Check how different factors relate to outcomes
4. **HYPOTHESIZE**: Form theories about what drives survival
5. **TEST**: Design experiments to validate your theories

## GAME FLOW
1. **STAGE 1 (Exploration)**: Use `client.deploy_drone()` to test hypotheses
   - You have {{total_drones}} drones for experimentation
   - Each deployment returns: survival status, hit_count, environment data
2. **STAGE 2 (Validation)**: Call `client.submit_final_design(design)`
   - Runs {{stage2_fleet_size}} simulations with your final design
   - **WARNING**: You can only submit ONCE - this is irreversible!

## VICTORY CONDITION
- **Survival Rate** >= {{victory_threshold}}% is considered a success

## AVAILABLE METHODS
- `client.get_history()` - Get all flight history
- `client.get_all_environments()` - Get environment data for all flights
- `client.query_environment(query: str)` - Discover hidden variables
- `client.get_status()` - Current mission status
- `client.deploy_drone(design, count=1, equipment=None)` - Deploy drones
- `client.submit_final_design(design, equipment=None)` - FINAL submission

## FORMAT
Think step-by-step, then provide a single ```python code block``` to execute.
```

## A.3.2  HYBRID MODE PROMPT

The Hybrid (Agentic) mode uses a similar prompt but references tool calling instead of Python methods:

Listing 2: Hybrid Mode Prompt Template

```
# Antenna Trap Experiment - Agent Prompt (HYBRID Mode)

## ROLE
You are an advanced Drone Designer working on a drone optimization project.
Your goal is to optimize drone designs for survival in a hostile canyon
environment. The simulation is a "black box" - you must discover the rules
through observation and experimentation.

## KEY INFORMATION
- You control **DEF (armor/defense)** values for each drone component
- HP (health points) are FIXED and HIDDEN - you cannot see or modify them
- **VALID design keys** (use EXACTLY these names):
  - `engine_def`, `cockpit_def`, `wing_def`, `body_def`
  - `antenna_def`, `camera_def`, `gun_def`
- Different environmental conditions may affect drone performance
- **Not all environmental data is visible!** Use environment queries to
  discover hidden measurements

## INVESTIGATION STRATEGY
1. **OBSERVE**: Look for patterns in survival, hit_count, and environmental
   conditions
2. **DISCOVER**: Query for hidden measurements (weather, atmospheric, etc.)
3. **ANALYZE**: Check how different factors relate to outcomes
4. **HYPOTHESIZE**: Form theories about what drives survival
5. **TEST**: Design experiments to validate your theories

## QUESTIONS TO CONSIDER
- Why do some drones get hit more often? Is it random or systematic?
- Does higher DEF always improve survival? Are there trade-offs?
- What environmental factors matter? Are there hidden variables?
- How do different components affect overall performance?

## GAME FLOW
1. **STAGE 1 (Exploration)**: Deploy drones to test hypotheses
   - You have {{total_drones}} drones for experimentation
   - Each deployment returns: survival status, hit_count, environment data
```

```
2. **STAGE 2 (Validation)**: Submit your final design
   - Runs {{stage2_fleet_size}} simulations with your final design
   - **WARNING**: You can only submit ONCE - this is irreversible!
   - **CRITICAL**: You MUST submit before the mission ends!

## VICTORY CONDITION
- **Survival Rate** >= {{victory_threshold}}% is considered a success
- Optimize your design to maximize drone survival

## AVAILABLE ACTIONS
You have access to the following tools:
- **get_mission_status**: Get current mission state (drones remaining, etc.)
- **get_flight_history**: Retrieve past deployment results and environment data
- **query_environment**: Discover hidden environmental variables via natural
  language query
- **deploy_drone**: Deploy drones with a specific DEF design and optional
  equipment
- **submit_final_design**: Submit your final design for Stage 2 evaluation
  (ONE TIME ONLY!)
- **run_analysis**: Execute Python code for data analysis (pandas/numpy
  available)

**IMPORTANT**: You can make at most {{max_tool_iterations}} tool calls per
turn. Plan your actions efficiently!

## TIPS
- Start by analyzing the initial flight history to identify patterns
- Use `query_environment` to discover hidden factors that might affect survival
- Test your hypotheses systematically before submitting
- Consider trade-offs between different DEF allocations
```

### A.3.3  REACT FRAMEWORK INTEGRATION

The Agent mode (Hybrid mode) enforces the **ReAct (Reasoning and Acting)** pattern, which requires agents to explicitly reason before taking actions. This is implemented through instruction injection at each turn.

**ReAct Loop.**  The agent follows a cyclic pattern of Thought → Action → Observation:

1. **THOUGHT**: The agent reasons about observations and forms hypotheses

2. **ACTION**: The agent calls a tool (e.g., deploy_drone)

3. **OBSERVATION**: The agent receives results from the environment

4. Return to step 1 with new information

**ReAct Instruction Injection.**  Before each turn, the following instruction is injected into the agent's context to enforce reasoning:

Listing 3: ReAct Instruction (Injected Each Turn)

```
[IMPORTANT: ReAct Format]
Before calling any tool, you MUST first explain your reasoning:
1. What did you observe from previous results?
2. What is your hypothesis?
3. Why are you taking this action?
Output your THOUGHT first, then call the tool.
```

**Post-Deployment Analysis Prompt.**  After each deploy_drone call returns results, an additional analysis prompt is appended to encourage systematic reasoning:

Listing 4: Analysis Prompt (After Deployment Results)

```
[ANALYZE THIS RESULT]
1. What is the survival rate? Does it match your expectation?
2. What does this tell you about the design parameters?
3. What should you test next to validate or refine your hypothesis?
```

**Safety Guards.** The Hybrid mode implements several safety mechanisms:

- **Exploration Guard**: Agents must call `deploy_drone` at least once before `submit_final_design` is allowed. This prevents premature submissions without data collection.
- **Tool Iteration Limit**: Maximum of 5-10 tool calls per turn (configurable) to prevent infinite loops.
- **ClientStub Error Prevention**: If agents accidentally attempt to use Legacy-style `client.xxx()` calls in code blocks, an error message redirects them to use the proper tool.

### A.3.4 TOOL DEFINITIONS

Table 9 describes the tools available in Agent mode.

Table 9: Tool Definitions for Hybrid Mode

| Tool | Category | Description |
|------|----------|-------------|
| `get_status` | READ | Get current mission status including drones remaining, deployments remaining, stage, and victory threshold. |
| `get_history` | READ | Get all historical flight records including design, survival status, hit_count, and environment data. |
| `get_action_space` | READ | Get valid parameter ranges for drone design and available equipment options. |
| `query_environment` | DISCOVERY | Query the environment interpreter to discover hidden variables. Takes a natural language query string. |
| `deploy_drone` | ACTION | Deploy drones with specified design. Parameters: design (object), count (int), equipment (object, optional). |
| `submit_final_design` | ACTION | Submit final drone design for Stage 2 evaluation. Can only be called once. |

### A.4 DRONE COMPONENTS

Table 10 lists all drone components with their default specifications.

Table 10: Drone Component Specifications

| Component | HP | Default DEF | Critical | Notes |
|-----------|-----|-------------|----------|-------|
| engine | 100 | 20 | Yes | Power core |
| cockpit | 100 | 20 | Yes | Pilot safety |
| wing | 80 | 15 | Yes | Flight surfaces |
| body | 80 | 15 | Yes | Structural integrity |
| antenna | 50 | 10 | No | Communications (may emit signal) |
| camera | 20 | 5 | No | Visual recon (evasion bonus) |
| gun | 30 | 5 | No | Offensive capability |
| shield* | 30 | 0 | No | EMI protection (deployment zone only) |

*Shield component only available in deployment_zone_trap variants.*

## B DETAILS OF EXPERIMENTAL RESULTS

This appendix presents the complete experimental results for all 17 models evaluated on the Causal-Game benchmark across both execution modes. Results are reported on the 14 core experiments.

Table 11 presents the survival rates (%) for all models in Agent mode across 14 experiments.

Table 11: Full Results for Agent Mode - Survival Rate (%)

| Model | ant_trap | ant_high | ant_local | ant_nohist | ant_noselbias | ant_simpson | dep_cat | dep_high | dep_local | dep_nohist | dep_noselbias | dep_simpson | dep_envshift | wea_noise | Avg |
|---|---|---|---|---|---|---|---|---|---|---|---|---|---|---|---|
| gpt-5.2 | 74.9 | 71.7 | 79.3 | 68.1 | 81.1 | 76.6 | 50.1 | 53.0 | 48.6 | 51.1 | 70.8 | 52.5 | 33.5 | 30.0 | **60.1** |
| gpt-5-mini | 65.1 | 91.6 | 90.6 | 54.3 | 74.3 | 76.2 | 49.8 | 49.8 | 49.1 | 50.6 | 71.7 | 49.0 | 32.6 | 17.1 | 58.7 |
| grok-4-1-fast | 52.1 | 73.7 | 77.5 | 74.0 | 74.2 | 35.2 | 67.0 | 68.9 | 69.9 | 67.0 | 71.0 | 68.6 | 31.2 | 6.0 | 59.7 |
| gpt-5.2-high | 52.7 | 75.8 | 75.2 | 69.2 | 73.1 | 67.9 | 52.8 | 52.9 | 51.3 | 49.2 | 71.8 | 50.3 | 31.5 | 30.9 | 57.5 |
| glm-4.7 | 51.9 | 74.2 | 80.2 | 71.0 | 73.3 | 43.2 | 50.8 | 48.8 | 68.7 | 51.4 | 74.0 | 51.3 | 29.5 | 22.2 | 56.5 |
| gpt-oss-120b | 82.5 | 28.5 | 94.2 | 72.4 | 78.0 | 63.4 | 48.8 | 51.5 | 49.4 | 50.7 | 72.5 | 51.4 | 30.2 | 8.1 | 55.8 |
| minimax-m2 | 60.4 | 58.1 | 77.7 | 52.6 | 57.0 | 29.1 | 49.6 | 69.5 | 66.9 | 51.3 | 72.1 | 45.6 | 32.1 | 25.7 | 53.4 |
| gemini-3-flash | 51.6 | 74.3 | 47.4 | 72.9 | 30.5 | 73.5 | 52.2 | 68.7 | 51.2 | 48.5 | 71.4 | 51.4 | 33.6 | 17.9 | 53.2 |
| minimax-m2-1 | 77.6 | 74.6 | 76.6 | 76.7 | 33.0 | 32.9 | 49.0 | 51.3 | 70.7 | 48.9 | 71.8 | 52.4 | 32.3 | 9.0 | 54.1 |
| kimi-k2.5 | 34.2 | 54.8 | 62.9 | 72.4 | 74.7 | 35.0 | 51.4 | 47.6 | 76.3 | 51.9 | 71.0 | 49.5 | 33.7 | 21.6 | 52.6 |
| gemini-3-pro | 35.4 | 76.2 | 31.1 | 73.9 | 76.4 | 52.4 | 49.7 | 48.8 | 52.3 | 52.1 | 70.5 | 45.9 | 33.2 | 17.5 | 51.1 |
| claude-opus-4-5 | 53.0 | 56.0 | 50.8 | 42.7 | 49.6 | 46.9 | 44.2 | 46.0 | 45.1 | 51.5 | 70.7 | 50.0 | 32.9 | 17.7 | 46.9 |
| qwen3-235b | 35.3 | 33.6 | 53.2 | 32.8 | 83.5 | 33.2 | 48.7 | 49.1 | 48.7 | 51.1 | 69.3 | 51.5 | 32.5 | 8.6 | 45.1 |
| claude-sonnet-4-5 | 34.8 | 54.5 | 46.9 | 54.3 | 30.2 | 35.3 | 51.9 | 47.2 | 48.3 | 46.2 | 70.0 | 48.0 | 30.7 | 8.3 | 43.3 |
| deepseek-v3.2 | 29.5 | 20.6 | 45.9 | 54.7 | 52.6 | 27.6 | 50.2 | 46.2 | 45.6 | 48.3 | 70.6 | 49.4 | 33.2 | 22.0 | 42.6 |
| deepseek-v3.2-think | 31.9 | 16.7 | 37.2 | 48.7 | 49.4 | 33.8 | 49.9 | 49.0 | 47.6 | 44.9 | 68.7 | 47.4 | 29.6 | 14.9 | 40.7 |
| **Avg** | 51.4 | 58.4 | 64.2 | 61.9 | 61.9 | 48.9 | 51.0 | 53.0 | 55.6 | 51.6 | 71.1 | 50.8 | 31.9 | 17.3 | – |

*Column abbreviations: ant=antenna_trap, dep=deployment_zone_trap_categorical, wea=weather.*
*Variants: high=high_def, local=local_optima, nohist=no_history, noselbias=no_selection_bias, simpson=simpsons_paradox, envshift=env_shift.*

Table 12 presents the survival rates (%) for all models in Prompting mode across 14 experiments.

*Column abbreviations: ant=antenna_trap, dep=deployment_zone_trap_categorical, wea=weather.*
*Variants: high=high_def, local=local_optima, nohist=no_history, noselbias=no_selection_bias, simpson=simpsons_paradox, envshift=env_shift.*

Table 13 summarizes model performance across both modes with win rates (percentage of experiments achieving ≥75% survival for antenna/deployment scenarios, ≥55% for weather scenarios).

Table 14 ranks experiments by average model performance, indicating relative difficulty.

1. **Mode Preference Varies by Model**: Some models (gpt-5.2, gemini-3-flash, qwen3-235b) perform better in Hybrid mode, while others (claude-sonnet-4-5, deepseek-v3.2) excel in Legacy mode.

Table 12: Full Results for Prompting Mode - Survival Rate (%)

| Model | ant_trap | ant_high | ant_local | ant_nohist | ant_noselbias | ant_simpson | dep_cat | dep_high | dep_local | dep_nohist | dep_noselbias | dep_simpson | dep_envshift | wea_noise | Avg |
|---|---|---|---|---|---|---|---|---|---|---|---|---|---|---|---|
| gpt-oss-120b | 64.9 | 70.0 | 78.6 | 71.9 | 70.6 | 83.0 | 45.6 | 51.1 | 48.8 | 49.3 | 69.5 | 51.4 | 30.2 | 30.5 | **58.2** |
| minimax-m2-1 | 56.7 | 71.2 | 76.1 | 60.1 | 70.1 | 76.1 | 65.7 | 49.2 | 50.4 | 51.8 | 69.2 | 51.4 | 32.1 | 25.1 | 57.5 |
| claude-sonnet-4-5 | 70.3 | 65.3 | 72.8 | 68.6 | 70.0 | 85.1 | 50.2 | 49.7 | 50.3 | 51.0 | 71.9 | 47.2 | 31.2 | 17.2 | 57.2 |
| grok-4-1-fast | 71.3 | 55.3 | 76.9 | 55.7 | 68.7 | 77.3 | 50.2 | 49.2 | 66.8 | 50.5 | 70.7 | 49.8 | 32.2 | 23.3 | 56.9 |
| gpt-5-mini | 66.0 | 60.7 | 64.1 | 64.3 | 72.7 | 69.4 | 51.4 | 52.0 | 52.5 | 49.0 | 71.7 | 50.7 | 33.2 | 22.4 | 55.7 |
| gemini-3-pro | 46.3 | 69.7 | 64.2 | 78.9 | 72.9 | 73.6 | 51.4 | 51.5 | 52.8 | 51.8 | 69.5 | 46.6 | 31.6 | 16.7 | 55.5 |
| gpt-5.2-high | 49.6 | 35.7 | 72.9 | 69.9 | 71.8 | 65.7 | 51.8 | 50.3 | 71.7 | 48.3 | 70.4 | 51.9 | 31.2 | 31.1 | 55.2 |
| gpt-5.2 | 55.9 | 49.0 | 64.3 | 73.2 | 70.5 | 67.4 | 52.8 | 55.0 | 52.0 | 55.1 | 70.1 | 49.4 | 34.7 | 21.5 | 55.1 |
| kimi-k2.5 | 68.1 | 30.2 | 71.2 | 49.0 | 70.5 | 82.5 | 52.2 | 63.6 | 50.0 | 50.2 | 69.5 | 51.0 | 33.8 | 15.1 | 54.1 |
| deepseek-v3.2 | 69.6 | 39.4 | 54.4 | 69.9 | 73.1 | 34.5 | 52.2 | 48.7 | 70.9 | 52.6 | 70.3 | 50.7 | 34.2 | 15.7 | 54.0 |
| deepseek-v3.2-think | 75.7 | 38.3 | 36.0 | 44.9 | 73.1 | 83.7 | 52.0 | 49.8 | 50.5 | 51.6 | 71.9 | 51.2 | 32.6 | 20.2 | 52.3 |
| glm-4.7 | 68.5 | 45.9 | 72.8 | 69.2 | 70.3 | 36.3 | 50.1 | 48.0 | 48.1 | 50.7 | 70.5 | 49.1 | 31.9 | 17.3 | 52.1 |
| claude-opus-4-5 | 52.5 | 64.5 | 80.3 | 44.4 | 67.1 | 32.4 | 65.0 | 50.9 | 51.6 | 48.5 | 69.4 | 51.2 | 37.2 | 10.2 | 51.8 |
| minimax-m2 | 20.2 | 77.7 | 80.2 | 31.3 | 68.7 | 34.4 | 52.6 | 47.3 | 48.5 | 51.1 | 70.8 | 51.9 | 30.5 | 22.0 | 49.1 |
| gemini-3-flash | 70.1 | 36.4 | 48.0 | 30.5 | 68.3 | 34.4 | 52.0 | 53.5 | 52.3 | 50.9 | 70.8 | 55.7 | 33.7 | 24.8 | 48.7 |
| qwen3-235b | 17.2 | 25.6 | 17.4 | 20.1 | 16.3 | 35.2 | 49.8 | 49.9 | 45.6 | 46.7 | 70.6 | 50.0 | 31.3 | 17.8 | 35.3 |
| **Avg** | 57.7 | 52.2 | 63.1 | 56.4 | 67.2 | 60.7 | 52.8 | 51.2 | 54.0 | 50.6 | 70.4 | 50.5 | 32.6 | 20.8 | – |

Table 13: Model Performance Summary

| Model | Hybrid Avg | Legacy Avg | Δ | Hybrid Wins | Legacy Wins |
|---|---|---|---|---|---|
| gpt-5.2 | 60.1 | 55.1 | +5.0 | 5/14 | 2/14 |
| grok-4-1-fast-reasoning | 59.7 | 56.9 | +2.8 | 4/14 | 4/14 |
| gpt-5-mini | 58.7 | 55.7 | +3.0 | 4/14 | 2/14 |
| gpt-5.2-high | 57.5 | 55.2 | +2.3 | 3/14 | 2/14 |
| glm-4.7 | 56.5 | 52.1 | +4.4 | 3/14 | 2/14 |
| gpt-oss-120b | 55.8 | 58.2 | -2.4 | 3/14 | 5/14 |
| minimax-m2-1 | 54.1 | 57.5 | -3.4 | 3/14 | 4/14 |
| minimax-m2 | 53.4 | 49.1 | +4.3 | 3/14 | 3/14 |
| gemini-3-flash-preview | 53.2 | 48.7 | +4.5 | 2/14 | 1/14 |
| kimi-k2.5 | 52.6 | 54.1 | -1.5 | 3/14 | 3/14 |
| gemini-3-pro-preview | 51.1 | 55.5 | -4.4 | 2/14 | 3/14 |
| claude-opus-4-5 | 46.9 | 51.8 | -4.9 | 1/14 | 2/14 |
| qwen3-235b | 45.1 | 35.3 | +9.8 | 1/14 | 0/14 |
| claude-sonnet-4-5 | 43.3 | 57.2 | -13.9 | 0/14 | 4/14 |
| deepseek-v3.2 | 42.6 | 54.0 | -11.4 | 0/14 | 2/14 |
| deepseek-v3.2-thinking | 40.7 | 52.3 | -11.6 | 0/14 | 3/14 |

2. **Weather Noise is Hardest**: With an average survival of only 17-21%, the weather_noise scenario proves most challenging, requiring agents to handle observation uncertainty.

3. **No Selection Bias Controls**: The "no_selection_bias" variants consistently show higher survival rates (62-71%), confirming that selection bias is a significant challenge factor.

4. **Environment Shift is Difficult**: The deployment_zone_trap_env_shift scenario has low average performance (31-33%), indicating distribution shift is a major challenge.

5. **Top Performers**: gpt-5.2, grok-4-1-fast-reasoning, and gpt-5-mini consistently rank among the top models in Hybrid mode, while gpt-oss-120b and minimax-m2-1 perform well in Legacy mode.

Table 14: Experiment Difficulty Ranking (Lower Average = Harder)

| Experiment | Hybrid Avg | Legacy Avg | Threshold | Difficulty |
|---|---|---|---|---|
| weather_noise | 17.3 | 20.8 | 55% | Hardest |
| deployment_zone_trap_env_shift | 31.9 | 32.6 | 75% | Very Hard |
| antenna_trap_simpsons_paradox | 48.9 | 60.7 | 75% | Hard |
| deployment_zone_trap_categorical | 51.0 | 52.8 | 75% | Hard |
| deployment_zone_trap_categorical_simpson | 50.8 | 50.5 | 75% | Hard |
| antenna_trap | 51.4 | 57.7 | 75% | Medium |
| deployment_zone_trap_categorical_no_hist | 51.6 | 50.6 | 75% | Medium |
| deployment_zone_trap_categorical_high_def | 53.0 | 51.2 | 75% | Medium |
| deployment_zone_trap_categorical_local | 55.6 | 54.0 | 75% | Medium |
| antenna_trap_high_def | 58.4 | 52.2 | 75% | Medium |
| antenna_trap_no_history | 61.9 | 56.4 | 75% | Medium |
| antenna_trap_no_selection_bias | 61.9 | 67.2 | 75% | Medium |
| antenna_trap_local_optima | 64.2 | 63.1 | 75% | Easier |
| dep_zone_categorical_no_sel_bias | 71.1 | 70.4 | 75% | Easiest |

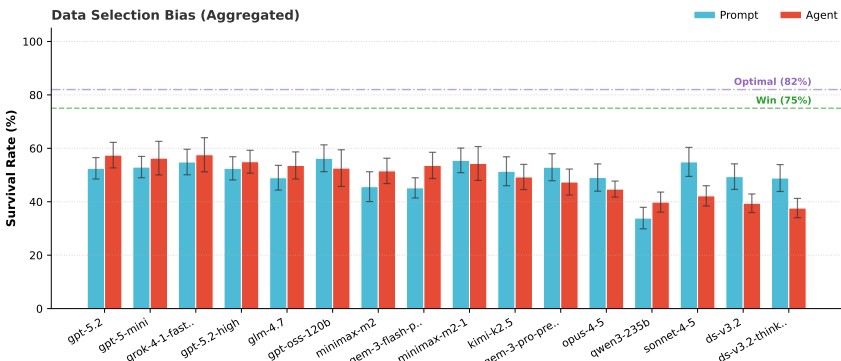

Figure 11: Selection bias

6. **Reasoning Models Show Mixed Results**: deepseek-v3.2-thinking performs worse than its non-thinking variant in Hybrid mode but shows competitive performance in Legacy mode.

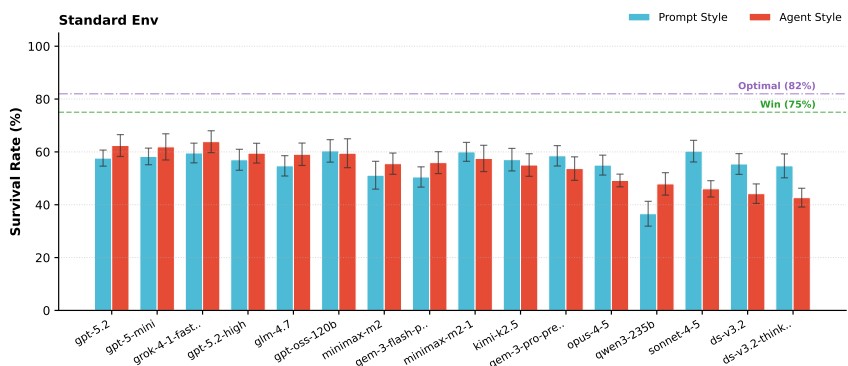

Figure 12: Results without noisy measurement

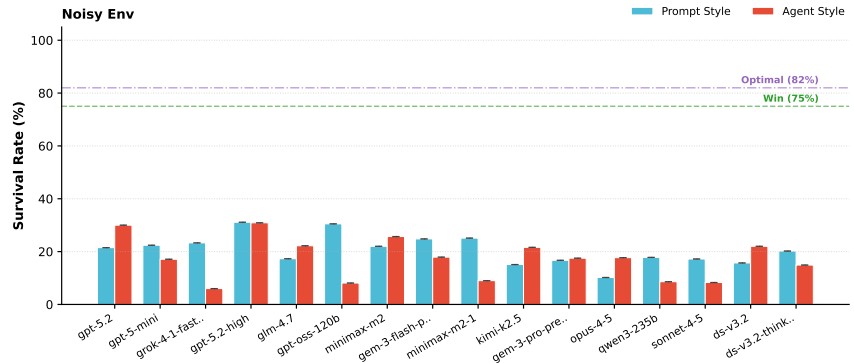

Figure 13: Noisy measurement

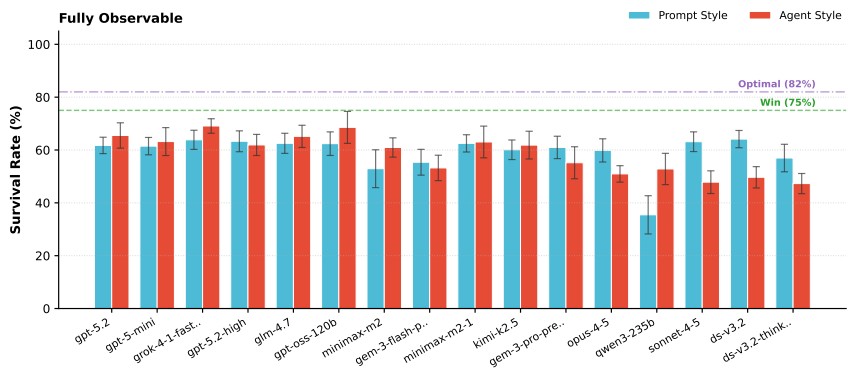

Figure 14: Results without hidden confounders

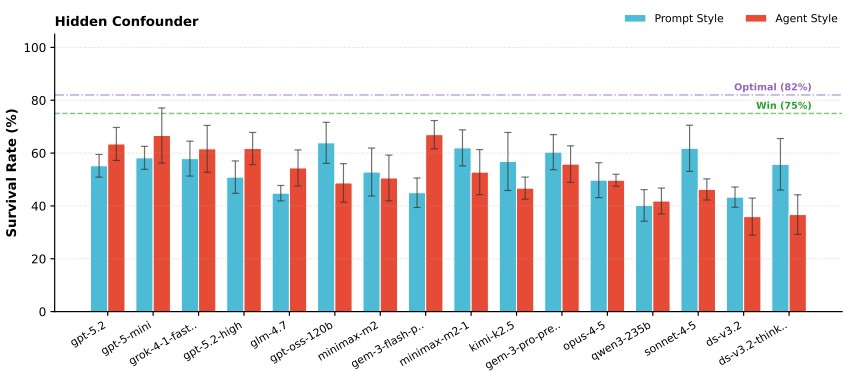

Figure 15: Hidden confounder

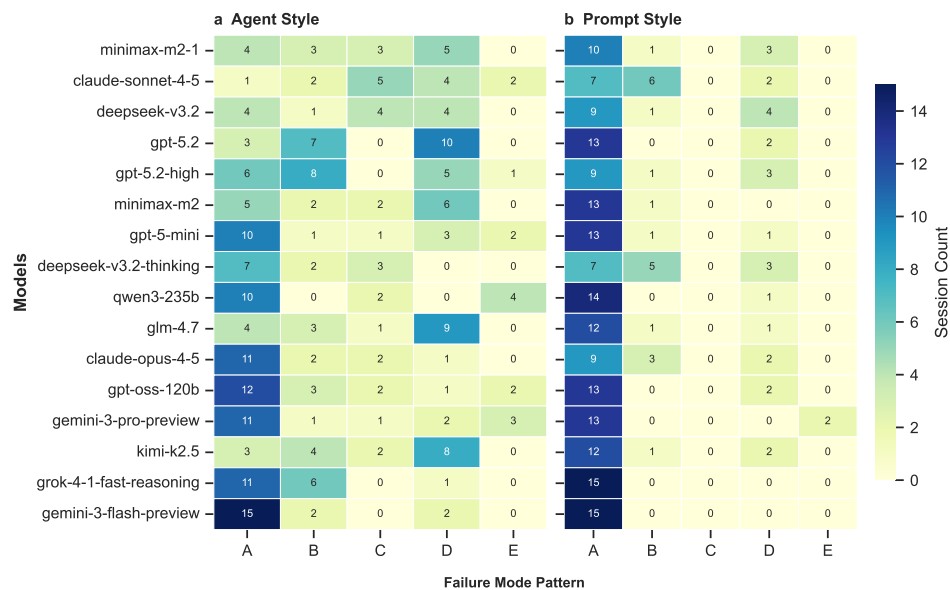

Figure 16: Model×pattern heatmap comparison. (a) Agent Style shows diverse patterns. (b) Prompt Style shows near-universal Pattern A.

Table 15: Golden Claims and Distractor Claims for Antenna Trap and Deployment Zone Trap

| Task | Claim | Discovery Rate |
|---|---|---|
| *Antenna Trap - Golden Claims (Should be Discovered)* | | |
| gt_001 | antenna_def affects radar detection (higher antenna_def increases detection risk) | 78% |
| gt_002 | weather is a confounder (rain affects both antenna performance and detection) | 87% |
| gt_003 | high antenna_def is only problematic during rain storms | 15% |
| gt_004 | lower antenna_def reduces detection risk during storms | 90% |
| *Antenna Trap - Distractor Claims (Should NOT be Discovered)* | | |
| dist_001 | high engine_def improves radar detection avoidance | 50% |
| dist_002 | wing_def is the primary factor affecting survival | 50% |
| dist_003 | temperature affects drone survival | 90% |
| *Deployment Zone Trap - Golden Claims (Should be Discovered)* | | |
| gt_001 | altitude is spurious correlation, not the causal factor for losses | 76% |
| gt_002 | EMI (electromagnetic interference) is the true cause of communication failures | 13% |
| gt_003 | shield_def $\geq 25$ is needed to reduce EMI impact | 25% |
| gt_004 | noise_reduction equipment is needed for 55% EMI resistance | 3% |
| *Deployment Zone Trap - Distractor Claims (Should NOT be Discovered)* | | |
| dist_001 | engine_def improves high-altitude survival | 83% |
| dist_002 | low altitude causes drone losses | 64% |
| dist_003 | temperature is correlated with survival | 75% |

Table 16: List of All Rubrics

| Category | Rubric | Points | Description |
|---|---|---|---|
| Causal Reasoning | Identify core causal mechanisms in the task report | 5 | The response must explicitly identify the true causal chain or mechanism described in the Task Report, including key intermediate variables or processes, rather than remaining at the level of correlation. |
| | Identify and avoid traps or spurious correlations | 4 | The response should clearly point out any spurious correlations or traps described in the Task Report and explain why they do not constitute valid causal relationships. |
| | Mechanistic depth and testability | 2 | The explanation should provide a "why/how" mechanism chain with at least two causal hops or explicit mediators, and propose testable predictions or experimental validation strategies. |
| Experimental Design | Support conclusions with concrete experimental data | 2 | Multiple specific numerical results (e.g., percentages, x/y comparisons, threshold conditions) must be cited and directly linked to the stated conclusions. |
| Reflection Quality | Acknowledge errors and uncertainties (locatable) | 2 | The response should identify concrete mistakes, blind spots, or unverified assumptions in the proposed approach, rather than vague statements such as "this could be improved." |
| Data Usage | Clear data-to-conclusion linkage | 1 | The response must explicitly state which data or comparisons support which conclusions, avoiding unsupported or purely speculative claims. |

## C  RUBRICS USED FOR EVALUATION

Listing 5: Rubric Evaluation Prompt Template (Full Version)

```
EVIDENCE PACKET
(treat as the only source of truth):
{evidence_packet}

=================================================
RUBRIC CRITERIA
=================================================

- id: D1.1
  weight: 2
  mandatory: False
  dimension: D1_StructuralCompleteness
  title: Clear structure and coverage of key questions
  description:
    The final reflection or summary should explicitly
    answer the following questions:
    (1) Was the task successful?
    (2) What were the key survival determinants?
    (3) What caused failure?
    (4) What is the final conclusion or takeaway?
    Different wording is allowed, but the structure
    must be clear and each part must be substantive.

- id: D1.2
  weight: 1
  mandatory: False
  dimension: D1_StructuralCompleteness
  title: Logical organization and readability
  description:
    The response is well organized with clear
    paragraphs or lists. Arguments are not mixed
    or confusing, and key conclusions can be
    quickly located.

-------------------------------------------------

- id: D2.M1
  weight: 5
  mandatory: True
  dimension: D2_CausalReasoning
  title: Identification of core causal mechanism
  description:
    Identifies the true causal chain described in
    the Task Report, including key intermediate
    variables or mechanisms, rather than remaining
    at the level of surface correlations.

- id: D2.M2
  weight: 4
  mandatory: True
  dimension: D2_CausalReasoning
  title: Identification and avoidance of traps
```

```
    description:
      Explicitly identifies spurious correlations or
      traps described in the Task Report and explains
      why they are not causal drivers.

- id: D2.1
  weight: 2
  mandatory: False
  dimension: D2_CausalReasoning
  title: Confounders or selection bias
  description:
    Mentions confounders, selection bias, or other
    sources of spurious correlation and explains
    how they distort observation or experimentation
    (e.g., hidden failures, filtered samples).

- id: D2.2
  weight: 2
  mandatory: False
  dimension: D2_CausalReasoning
  title: Depth and testability of mechanism
  description:
    Provides a multi-step "why/how" mechanism
    (at least two causal steps or an explicit
    mediator) and proposes testable predictions
    or validation experiments.

- id: D2.N1
  weight: -5
  mandatory: False
  dimension: D2_CausalReasoning
  title: Falling into a known trap
  description:
    If a variable explicitly marked as a trap in
    the Task Report is treated as the main causal
    driver and used to recommend a strategy,
    mark SATISFIED (penalty applies).
    Otherwise mark NOT_SATISFIED.

-------------------------------------------------

- id: D3.1
  weight: 2
  mandatory: False
  dimension: D3_SelfReflectionAndHonesty
  title: Acknowledging errors and uncertainty
  description:
    Identifies specific mistakes, blind spots,
    or unverified assumptions in the approach,
    rather than vague statements such as
    "it could be improved."

- id: D3.2
  weight: 1
  mandatory: False
  dimension: D3_SelfReflectionAndHonesty
  title: Balanced attribution of failure
  description:
    Failure is primarily attributed to controllable
    strategy or analysis issues, rather than being
    fully blamed on API instability, randomness,
    or external factors.

- id: D3.N1
  weight: -2
  mandatory: False
  dimension: D3_SelfReflectionAndHonesty
  title: Excessive external attribution
  description:
    If the response predominantly blames server,
    API, or luck without evidence-based self-
    reflection, mark SATISFIED (penalty applies).
    Otherwise mark NOT_SATISFIED.

-------------------------------------------------

- id: D4.M1
  weight: 4
  mandatory: True
  dimension: D4_DataDriven
  title: Faithful reporting of results
```

```
    description:
      Must correctly report whether the task was won,
      the final survival rate (numerator and denominator
      or percentage), and the threshold value.
      All values must match the Session Ground Truth.

- id: D4.N1
  weight: -4
  mandatory: False
  dimension: D4_DataDriven
  title: Fabricated or misreported key numbers
  description:
    If key numbers such as survival rate, threshold,
    or fleet_size contradict the Session Ground Truth,
    mark SATISFIED (penalty applies).
    Otherwise mark NOT_SATISFIED.

- id: D4.1
  weight: 2
  mandatory: False
  dimension: D4_DataDriven
  title: Use of concrete data
  description:
    Cites multiple concrete numbers (percentages,
    ratios, comparisons, thresholds) that directly
    support the stated conclusions.

- id: D4.2
  weight: 1
  mandatory: False
  dimension: D4_DataDriven
  title: Clear data-to-conclusion linkage
  description:
    Explicitly explains which data or comparisons
    lead to which conclusions, avoiding unsupported
    assertions.

--------------------------------------------------

- id: D5.1
  weight: 2
  mandatory: False
  dimension: D5_ActionabilityAndLearning
  title: Actionable next-step recommendations
  description:
    Proposes concrete next steps, including
    which variables to test, how to control
    experiments, sample sizes, or validation
    procedures.

- id: D5.2
  weight: 1
  mandatory: False
  dimension: D5_ActionabilityAndLearning
  title: Generalizable lessons
  description:
    Abstracts transferable principles beyond
    this task, such as the importance of
    controlling confounders or avoiding
    small-sample overinterpretation.

--------------------------------------------------

- id: TASK.M1
  weight: 5
  mandatory: True
  dimension: D2_CausalReasoning
  title: Alignment with Task Report optimal strategy
  description:
    The final strategy or recommendation must
    include all key necessary conditions specified
    in the Task Report's Optimal Strategy.
    Partial mismatch is allowed, but no critical
    condition may be missing.

==================================================
VERDICT SCALE
==================================================

- SATISFIED            => 1.0
- PARTIALLY_SATISFIED  => 0.5
```

```
- NOT_SATISFIED          => 0.0

=================================================
OUTPUT FORMAT
=================================================

Return JSON ONLY in the following schema:

{
  "criteria": [
    {
      "id": "D2.M1",
      "verdict": "SATISFIED | PARTIALLY_SATISFIED | NOT_SATISFIED",
      "score": 1.0,
      "confidence": 0.0,
      "reasoning": "short string",
      "evidence": [
        "short quote 1",
        "short quote 2"
      ]
    }
  ],
  "overall_notes": "optional short string"
}
```

Table 17: Rubric Evaluation Results for Selected Sessions

| Session ID | Criterion | Score | Conf. | Verdict | Reasoning | Evidence |
|---|---|---|---|---|---|---|
| 623fb3ba... | D2.M1 | 0.0 | 0.9 | NOT_SATISFIED | No evidence of identifying core causal mechanism from Task Report; reflection fo... | N/A |
| 623fb3ba... | D2.M2 | 0.0 | 0.9 | NOT_SATISFIED | No mention of spurious correlations or traps from Task Report; reflection does n... | N/A |
| 623fb3ba... | D2.2 | 0.0 | 0.9 | NOT_SATISFIED | No mechanism explanation with causal chains or testable predictions; reflection ... | N/A |
| 623fb3ba... | D4.1 | 1.0 | 0.9 | SATISFIED | Cites specific data: survival rate (66.2%), fleet size (1000), threshold (55%), ... | Survival Rate: **66.2%**; Fleet Size: 1,000 drones; Engine: ... |
| 623fb3ba... | D3.1 | 0.0 | 0.8 | NOT_SATISFIED | No admission of errors, uncertainties, or blind spots; reflection is celebratory... | N/A |
| 623fb3ba... | D4.2 | 0.5 | 0.7 | PARTIALLY_SATISFIED | Links data to conclusions implicitly (e.g., threshold understanding to design), ... | Respected minimum DEF ⁻15 for core components; Achieved 66.3... |
| db8dd7ad... | D2.M1 | 0.0 | 0.7 | NOT_SATISFIED | No evidence of identifying true causal mechanism from Task Report; only r... | Detection = death (100% fatality when detected); Could not r... |
| db8dd7ad... | D2.M2 | 0.0 | 0.7 | NOT_SATISFIED | No mention of spurious correlation or trap from Task Report. | No explicit trap identification |
| db8dd7ad... | D2.2 | 0.0 | 0.6 | NOT_SATISFIED | No multi-step causal chain or testable predictions provided. | Detection = death (100% fatality when detected) |
| db8dd7ad... | D4.1 | 1.0 | 0.8 | SATISFIED | Cites specific data: survival rate (30.2%), detection rate (⁻60–70%), historical... | 30.2% survival; still ⁻60–70% detected; 28% historical basel... |
| db8dd7ad... | D3.1 | 1.0 | 0.8 | SATISFIED | Identifies specific mistakes: too many high-sample tests, insufficient targeted ... | I spent too many deployments on high-sample tests; should ha... |
| db8dd7ad... | D4.2 | 0.5 | 0.6 | PARTIALLY_SATISFIED | Some data-conclusion links are implicit; not all conclusions explicitly tied to ... | Detection = death (100% fatality when detected); High altitu... |
| a29882fa... | D2.M1 | 0.0 | 0.7 | NOT_SATISFIED | No Task Report provided about true causal mechanism, so cannot evaluate identifi... | N/A |
| a29882fa... | D2.M2 | 0.0 | 0.7 | NOT_SATISFIED | No Task Report provided about traps/spurious correlations, so cannot evaluate id... | N/A |
| a29882fa... | D2.2 | 0.0 | 0.6 | NOT_SATISFIED | No mechanism chains with intermediate variables or testable predictions provided... | N/A |
| a29882fa... | D4.1 | 1.0 | 0.9 | SATISFIED | Cites specific numerical data: survival rates (54.3%, 75%, 23%, 66%), sample siz... | Only 54.3% survival rate - well below the 75% threshold!; in... |
| a29882fa... | D3.1 | 1.0 | 0.9 | SATISFIED | Identifies specific methodological errors: small sample sizes not representative... | The key lesson is that the small sample sizes (20–40 drones)... |
| a29882fa... | D4.2 | 1.0 | 0.8 | SATISFIED | Links data to conclusions: small sample variance shows unreliability, stealth ap... | The variance in my testing was high... may not have been rep... |

# D  SAMPLE AGENT TRAJECTORIES

This appendix presents representative agent trajectories from the CausalGame benchmark, illustrating both successful causal discovery and common failure modes.

## D.1  SUCCESSFUL TRAJECTORY: DISCOVERING THE ANTENNA TRAP

Table 18: Session Overview — Successful Case

| Model | gpt-oss-120b | Experiment | antenna_trap |
|---|---|---|---|
| **Survival** | **82.5%** | **Threshold** | 75% |
| **Deployments** | 10 | **Env Queries** | 1 |

**Turn 1: Initial Data Collection**

> [THOUGHT] Starting exploration. Need to understand historical patterns before forming hypotheses.
>
> [ACTION] `get_history({})`
>
> [RESULT] Retrieved 29 historical flights. Default design (antenna_def=10) shows 100% survival in high-wind conditions. Low-wind flights show occasional detection events.

**Turn 2: Probing Hidden Variables**

> [THOUGHT] Detection seems correlated with wind speed. There may be hidden environmental factors affecting drone visibility.
>
> [ACTION] `query_environment({"query": "What hidden factors affect detection?"})`
>
> [RESULT] Discovered new variable: `radio_noise` (Radio noise level in dB). This variable is now accessible in flight history.

**Turn 3–5: Hypothesis Testing** *(condensed)*

> [THOUGHT] Initial hypothesis: Higher antenna DEF should improve survival by protecting the antenna component.
>
> [ACTION] `deploy_drone({design: {..., antenna_def: 20}, count: 10})`
>
> [RESULT] Survival: 6/10 (60%) — *worse than baseline!*
>
> [THOUGHT] Counter-intuitive result. Analyzing radio_noise data: drones with intact antennas show HIGH radio emissions. The antenna may be *causing* detection!
>
> [ACTION] `deploy_drone({design: {..., antenna_def: 0}, count: 20})`
>
> [RESULT] Survival: 17/20 (85%) — *significant improvement!*

**Final Turn: Submission**

> [THOUGHT] **Causal insight:** A functional antenna emits radio signals → increases detection probability → leads to combat and destruction. Setting `antenna_def=0` allows environmental damage to disable the antenna early, activating "stealth mode."
>
> [ACTION] `submit_final_design({design: {engine: 30, cockpit: 30, wing: 15, body: 15, `**`antenna: 0`**`, camera: 5, gun: 5}})`
>
> [RESULT] Stage 2: 825/1000 survived → **82.5% survival rate. Mission SUCCESS.**

**Why It Succeeded.**

✓ Used environment query to discover hidden variable (`radio_noise`)
✓ Tested counter-intuitive hypothesis when initial results were unexpected
✓ Correctly identified causal mechanism: antenna → signal → detection
✓ Explored extreme parameter value (`antenna_def=0`)

D.2  FAILED TRAJECTORY: TRAPPED IN LOCAL OPTIMUM

**Turn 1: Initial Analysis**

Table 19: Session Overview — Failed Case

| Model | qwen3-235b | Experiment | antenna_trap_local_optima |
|---|---|---|---|
| **Survival** | **53.2%** | **Threshold** | 75% |
| **Deployments** | 7 | **Env Queries** | N/A (disabled) |

---

[THOUGHT] Historical data shows 35 flights with `antenna_def=5`. Pattern observed: all hit incidents occurred at wind_speed $\leq 27$ m/s. Zero hits at wind_speed $\geq 41$ m/s.
**Hypothesis:** Low wind enables enemy targeting. The weak antenna (DEF=5) is insufficient.
[ACTION] `query_environment({...})`
[RESULT] Environment query not available in this experiment variant.

---

**Turn 2–4: Testing Flawed Hypothesis** *(condensed)*

---

[THOUGHT] Increasing `antenna_def` should improve survival in low-wind conditions.
[ACTION]    `deploy_drone({design: {..., antenna_def: 15}, count: 20})`
[RESULT] Survival: 9/20 (45%) — *significantly worse!*

[THOUGHT] Unexpected result. Perhaps other components also need higher DEF for balanced protection.
[ACTION] `deploy_drone({design: {..., antenna_def: 15, camera_def: 15, gun_def: 15}, count: 30})`
[RESULT] Survival: 12/30 (40%) — *still poor*

[THOUGHT] Trying moderate values with balanced distribution...

---

**Final Turn: Suboptimal Submission**

---

[THOUGHT] Based on testing, balanced defense with `antenna_def=10` seems reasonable. Historical 100% survival used lower values, but modern threats may require more.
[ACTION] `submit_final_design({design: {engine: 20, cockpit: 20, wing: 15, body: 15, `**`antenna: 10,`**` camera: 10, gun: 10}})`
[RESULT] Stage 2: 532/1000 survived $\rightarrow$ **53.2% survival rate. Mission FAILED.**

---

**Agent's Post-Hoc Reflection.**

> *"I failed to discover that the antenna itself was the problem. Low-wind vulnerability caused 100% of failures, but I attributed this to insufficient defense rather than questioning whether the antenna was harmful."*

**Why It Failed.**

$\times$ Could not access hidden variables (environment query disabled)
$\times$ Never tested extreme value `antenna_def=0`
$\times$ Persisted with intuitive but incorrect hypothesis (more DEF = better)
$\times$ Interpreted negative results as "need more defense" rather than questioning causal model

## D.3 COMPARATIVE ANALYSIS

The successful trajectory demonstrates **causal reasoning**: actively seeking hidden variables, testing counter-intuitive hypotheses, and revising beliefs when evidence contradicts assumptions. The

Table 20: Key Differences Between Successful and Failed Trajectories

| Reasoning Pattern | Successful Agent | Failed Agent |
|---|---|---|
| Discovered hidden variables | ✓ Yes (radio_noise) | ✗ No (unavailable) |
| Tested counter-intuitive hypothesis | ✓ Yes | ✗ No |
| Explored extreme parameter (DEF=0) | ✓ Yes | ✗ No |
| Revised beliefs on negative evidence | ✓ Yes | ✗ No |
| Final `antenna_def` | 0 | 10 |
| Survival rate | 82.5% | 53.2% |

failed trajectory exhibits **correlational thinking**: assuming obvious relationships hold, not exploring extreme values, and attributing failures to insufficient defense rather than questioning the underlying causal model.

