# OpenReview forum: "CausalGame: Benchmarking Causal Thinking of LLM Agents in Games"
_ICLR.cc/2026/Workshop/FM4Science — ICLR 2026 Workshop FM4Science Poster_

### Official Review · Reviewer_hm4q · 2026-02-14
**CausalGame: A Necessary Reality Check for AI Scientists via Causal Benchmarking**

**Rating:** 7
**Confidence:** 4

**Review:**

### **1. Summary**
This paper addresses a critical bottleneck in the development of "AI Scientist" agents: the ability to perform causal thinking. While existing benchmarks focus on literature synthesis or statistical regression, CausalGame introduces a platform where agents must navigate hidden variables, selection bias, and measurement noise to uncover true data-generating mechanisms. By simulating scientific discovery as an interactive drone-design game governed by Structural Causal Models (SCMs), the authors provide a rigorous assessment of 16 frontier LLM agents. Across 14 carefully designed scenarios and 16 frontier LLMs, the authors show that even state-of-the-art reasoning and agentic models consistently fail to recover correct causal structures, despite sometimes producing fluent explanations or high rubric scores. The work argues convincingly that causal thinking is a missing core capability of current AI Scientist agents and that existing benchmarks fail to measure it.

### **2.Evaluation of Quality, Clarity, Originality, and Significance**
**Quality:** The experimental design is exceptionally robust. It utilizes a dual-evaluation system: quantitative performance (fleet survival rate) and qualitative fine-grained analysis (a 22-point rubric). The inclusion of 14 diverse game scenarios (e.g., Antenna Trap, Deployment Zone Trap) ensures the findings are not artifacts of a single task

**Clarity**: The paper is well-structured, clearly defining its motivation and the SCM-based engine. Figure 1 and Figure 2 effectively illustrate the conceptual and practical difference between a "naive" (correlational) agent and a "causal" agent.

**Originality**: This work is highly original in its shift from passive data science benchmarks to active interactive discovery. It is systematically incorporate hidden confounders and selection bias as core challenges for AI Scientists, mimicking real-world obstacles.

**Significance**: As LLM agents are increasingly trusted with research tasks, this paper serves as a vital warning. It highlights that without a paradigm shift toward causal reasoning, AI Scientists may propose "correct-sounding" but physically flawed or even dangerous conclusions.

### **3. Pros:**
- Novel, well-motivated benchmark focused on causal thinking.
- Realistic modeling of scientific discovery challenges.
- Strong experimental coverage across models and settings.
- Insightful failure-mode and hypothesis-distance analyses.
- Clear evidence that causal reasoning is distinct from other LLM capabilities.

### **4. Cons**
- Dependence on LLM-based rubric judges may introduce bias.
- Limited discussion of reproducibility cost and computational overhead.
- No baseline comparison to classical causal discovery algorithms or symbolic agents.

---

### Official Review · Reviewer_KyY4 · 2026-02-20
**CausalGame is a timely and rigorously designed benchmark to address the shortcomings of artificial intelligence that focuses solely on statistical correlation while neglecting causal reasoning. This paper challenges historical scientific paradoxes and uses a systems model-based (SCM) governance environment to provide a cool and necessary simulated testing scenario for the field. The research is significant in its finding that while state-of-the-art large language models perform well on other benchmarks, they default to pattern matching without the right inductive biases for causality.**

**Rating:** 8
**Confidence:** 5

**Review:**

Strong Points:
1. Originality: The use of a multi-round interactive causal discovery environment to replace traditional QA or code generation benchmarking is highly original.
2. Rigorous Evaluation: The "Failure Mode Analysis" (modes A-E) is particularly insightful, revealing that agents often perform well in the reflection phase but may fail in the execution phase (mode C).
3. Clarity: The paper is clearly structured, with a clear connection between historical causal challenges and simulated game scenarios.
4. Significance: It points to a major bottleneck in the vision of "AI scientists," showing that even with procedural experimentation, existing models are "hollow" in causality.


Weak Points:
1. While the drone design scenario is effective, it remains slightly disconnected from "real" scientific data (e.g., biological sequences). However, its underlying SCM logic is universal. It would be even better if discussions on applying this to real-world scenarios could be added.
2. It is interesting to observe that “thinking” models (such as DeepSeek-v3.2-thinking) sometimes perform worse in the agent framework than their non-thinking counterparts in direct hints, but a deeper qualitative analysis could be needed to understand why agent overhead confuses these models.

---

### Official Review · Reviewer_Wcos · 2026-02-22

**Rating:** 5
**Confidence:** 3

**Review:**

# Summary
The paper introduces CausalGame, a benchmark designed to evaluate the causal thinking capabilities of LLM-based AI Scientist agents. Unlike existing benchmarks that focus on executing research pipelines or statistical data analysis, CausalGame tests an agent's ability to navigate hidden confounders, selection biases, and measurement noise. CausalGame is framed as a multi-turn interactive game in which the agent must optimize drone designs for survival. The agent allocates defense budgets, deploys test batches, analyzes observational data governed by an underlying structural causal model, and submits a final design alongside an explanatory report.

# Strengths
* The motivation is strong. The paper addresses a critical problem in evaluating "AI Scientists." While many benchmarks test coding or literature synthesis, this work points out that scientific discovery requires distinguishing causation from correlation.

* The proposed benchmark and the corresponding evaluation metric seem well-defined.

# Weakness
* The citation format is inconsistent. In Section 3.2, the citations are not appropriately placed in the parentheses, making this section very distracting to read. The historical cases cited here did not help readers grasp the underlying principles of these challenges.
* What does "agent style" mean in Fig. 4? Does it mean ReAct? If yes, what would performing an action look like in the benchmark?
* The writing in the game scenarios is a bit unclear to me. Could the authors explain how the antenna trap is related to the deployment zone trap?
* Can traditional non-LLM causal discovery algorithms (such as PC or FCI), combined with an LLM, perform well if fed the same observational data?

I am willing to increase my score if the writing can be better polished.

---

### Official Review · Reviewer_VvhT · 2026-02-23
**The authors present CausalGame, an interactive benchmark designed to evaluate the causal reasoning capabilities of Large Language Model (LLM) agents in the context of scientific discovery. Motivated by the observation that existing AI scientist benchmarks largely ignore hidden variables, CausalGame utilizes Structural Causal Models (SCMs)  as the underlying engine to generate 14 game scenarios. These scenarios introduce realistic causal challenges, including selection bias, measurement noise, and latent confounders (e.g., Simpson's Paradox). The evaluation employs a dual-metric approach: a quantitative task success score (drone survival rate) and an LLM-as-a-judge rubric assessing the generated scientific reports. Testing 16 frontier LLMs reveals a systemic failure in causal thinking across both direct prompting and agentic (ReAct) frameworks, with agents frequently falling for spurious correlations.**

**Rating:** 7
**Confidence:** 4

**Review:**

Strengths

SCM-Grounded Engine: Using explicit Structural Causal Models to generate environments is a highly rigorous approach. It allows for the precise injection of specific causal phenomena (selection bias, measurement error) and guarantees that there is a definitive, mathematically sound ground-truth mechanism to be discovered.

Dual-Evaluation Methodology: The combination of an objective performance metric (fleet survival rate) with a fine-grained rubric (Causal Reasoning, Experimental Design, Reflection Quality, Data Usage) is excellent. The identification of "Pattern C" (agents that write highly reflective, high-scoring reports but completely fail the actual task) exposes a critical flaw in relying solely on LLM self-reflection for evaluation.

Comprehensive Benchmarking: Evaluating 16 state-of-the-art models across 14 variants provides a robust and valuable snapshot of the current frontier. The finding that smaller, distilled models occasionally outperform their larger counterparts on causal tasks is an interesting empirical contribution.


Weaknesses

The LLM-as-a-Judge Paradox: The paper relies on an LLM-based judge to evaluate the "Causal Reasoning" rubric of the agents' final reports. However, the core thesis of the paper is that all frontier LLMs systematically fail at causal reasoning. If state-of-the-art LLMs lack causal thinking, they are theoretically unqualified to accurately evaluate the causal reasoning depth of other models. The authors must validate the LLM judge's scores against human expert annotations to ensure the rubric scores are reliable.

Domain Abstraction Gap: The paper frames this as a benchmark for "AI Scientist agents," yet the scenarios (drone combat, antenna defense, enemy radar) map to military/gaming operations rather than scientific discovery (e.g., wet-lab biology, material science). While the underlying causal math is the same, LLMs heavily rely on domain priors. An LLM might fail to recognize a confounder in "drone armor" but easily recognize it in "drug dosage vs. patient recovery" due to its training distribution.

Lack of Human Baselines: It is difficult to contextualize an average survival rate of 40-60% without knowing how humans perform. Are these traps mathematically solvable within the limited deployment budget, or are they effectively "trick questions" that even human scientists would fail without prior domain knowledge?

ReAct Rigidity: In the Hybrid mode, the prompt forces a strict ReAct format. The observed performance drop in some advanced models (like Claude-3.5-Opus) when moved from Prompting to Hybrid mode might be a failure of instruction-following regarding the rigid ReAct template rather than a degradation in causal thinking.

Questions for the Authors

Judge Validation: What model was used as the LLM-as-a-judge for the rubric evaluation? Have you calculated the alignment/correlation between the LLM judge's scores and human expert evaluations on a subset of the reports?

Semantic Framing: Have you experimented with re-skinning the SCMs into a purely scientific domain (e.g., changing "Antenna Defense" to "Catalyst Concentration" and "Weather" to "Temperature") to see if semantic framing activates better causal priors in the LLMs?

Human Performance: Can you provide a baseline of human performance (e.g., computer science graduate students) on a subset of the Antenna Trap and Zone Trap scenarios to calibrate the difficulty of the benchmark?

---

### Decision · Program_Chairs · 2026-03-03

Accept (Poster)